# Improve Object Detection with Feature-based Knowledge Distillation: Towards Accurate and Efficient Detectors

**Linfeng Zhang**
Institute for Interdisciplinary Information
Sciences, Tsinghua University

**Kaisheng Ma** *
Institute for Interdisciplinary Information
Sciences, Tsinghua University

## Abstract

Knowledge distillation, in which a student model is trained to mimic a teacher model, has been proved as an effective technique for model compression and model accuracy boosting. However, most knowledge distillation methods, designed for image classification, have failed on more challenging tasks, such as object detection. In this paper, we suggest that the failure of knowledge distillation on object detection is mainly caused by two reasons: (1) the imbalance between pixels of foreground and background and (2) lack of distillation on the relation between different pixels. Observing the above reasons, we propose attention-guided distillation and non-local distillation to address the two problems, respectively. Attention-guided distillation is proposed to find the crucial pixels of foreground objects with attention mechanism and then make the students take more effort to learn their features. Non-local distillation is proposed to enable students to learn not only the feature of an individual pixel but also the relation between different pixels captured by non-local modules. Experiments show that our methods achieve excellent AP improvements on both one-stage and two-stage, both anchor-based and anchor-free detectors. For example, Faster RCNN (ResNet101 backbone) with our distillation achieves 43.9 AP on COCO2017, which is 4.1 higher than the baseline. Codes have been released on Github[†].

## 1 Introduction

Recently, excellent breakthrough in various domains has been achieved with the success of deep learning (Ronneberger et al., 2015; Devlin et al., 2018; Ren et al., 2015). However, the most advanced deep neural networks always consume a large amount of computation and memory, which has limited their deployment in edge devices such as self-driving cars and mobile phones. To address this problem, abundant techniques are proposed, including pruning (Han et al., 2016; Zhang et al., 2018; Liu et al., 2018; Frankle & Carbin, 2018), quantization (Nagel et al., 2019; Zhou et al., 2017), compact model design (Sandler et al., 2018; Howard et al., 2019; Ma et al., 2018; Iandola et al., 2016) and knowledge distillation (Hinton et al., 2014; Buciluǎ et al., 2006). Knowledge distillation, which is also known as teacher-student learning, aims to transfer the knowledge of an over-parameterized teacher to a lightweight student. Since the student is trained to mimic the logits or features of the teacher, the

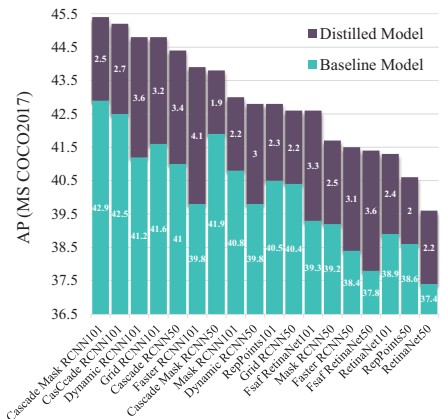

Figure 1: Results overview.

student can inherit the dark knowledge from the teacher, and thus often achieves much higher accuracy. Due to its simplicity and effectiveness, knowledge distillation has become a popular technique for both model compression and model accuracy boosting.

---

*Corresponding author

[†]https://github.com/ArchipLab-LinfengZhang/Object-Detection-Knowledge-Distillation-ICLR2021

As one of the most crucial challenges in computer vision, object detection has an urgent requirement of both accurate and efficient models. Unfortunately, most of the existing knowledge distillation methods in computer vision are designed for image classification and usually leads to trivial improvements on object detection (Li et al., 2017). In this paper, we impute the failure of knowledge distillation on object detection to the following two issues, which will be solved later, respectively.

**Imbalance between foreground and background.** In an image to be detected, the background pixels are often more overwhelming than the pixels of the foreground objects. However, in previous knowledge distillation, the student is always trained to mimic the features of all pixels with the same priority. As a result, students have paid most of their attention to learning background pixels features, which suppresses student's learning on features of the foreground objects. Since foreground pixels are more crucial in detection, the imbalance hurts the performance of knowledge distillation severely. To overcome this obstacle, we propose the attention-guided distillation which distills only the crucial foreground pixels. Since the attention map can reflect the position of the important pixels (Zhou et al., 2016), we adopt the attention map as the mask for knowledge distillation. Concretely, the pixel with a higher attention value is regarded as a pixel of a foreground object and then is learned by the student model with a higher priority. Compared with the previous binary mask method (Wang et al., 2019), the mask generated by attention maps in our methods is more fine-grained and requires no additional supervision. Compared with the previous attention-based distillation methods (Zagoruyko & Komodakis, 2017), the attention map in our methods is not only utilized as the information to be distilled but also utilized as the mask signal for feature distillation.

**Lack of distillation on relation information.** It is generally acknowledged that the relation between different objects contains valuable information in object detection. Recently, lots of researchers successfully improve the performance of detectors by enabling detectors to capture and make use of these relations, such as non-local modules (Wang et al., 2018) and relation networks (Hu et al., 2018). However, the existing object detection knowledge distillation methods only distill the information of individual pixels but ignore the relation of different pixels. To solve this issue, we propose the non-local distillation, which aims to capture the relation information of students and teachers with non-local modules and then distill them from teachers to students.

Since the non-local modules and attention mechanism in our methods are only required in the training period, our methods don't introduce additional computation and parameters in the inference period. Besides, our methods are feature-based distillation methods which do not depend on a specific detection algorithm so they can be directly utilized in all kinds of detectors without any modification. On MS COCO2017, 2.9, 2.9 and 2.2 AP improvements can be observed on two-stage, one-stage, and anchor-free models on average, respectively. Experiments on Mask RCNN show that our methods can also improve the performance of instance segmentation by 2.0 AP, on average. We have conducted a detailed ablation study and sensitivity study to show the effectiveness and stability of each distillation loss. Moreover, we study the relation between teachers and students on object detection and find that *knowledge distillation on object detection requires a high AP teacher, which is different from the conclusion in image classification where a high AP teacher may harm the performance of students* (Mirzadeh et al., 2019; Cho & Hariharan, 2019). We hope that these results are worth more contemplation of knowledge distillation on tasks except for image classification. To sum up, the contribution of this paper can be summarized as follows.

- We propose the attention-guided distillation, which emphasizes students' learning on the foreground objects and suppresses students' learning on the background pixels.
- We propose the non-local distillation, which enables the students to learn not only the information of the individual pixel but also the relation between different pixels from teachers.
- We show that a teacher with higher AP is usually a better teacher in knowledge distillation on object detection, which is different from the conclusion in image classification.

## 2 RELATED WORK

As an effective method for model compression and model accuracy boosting, knowledge distillation has been widely utilized in various domains and tasks, including image classification (Hinton et al., 2014; Romero et al., 2015; Zagoruyko & Komodakis, 2017), object detection (Chen et al., 2017; Li et al., 2017; Wang et al., 2019; Bajestani & Yang, 2020), semantic segmentation (Liu et al., 2019),

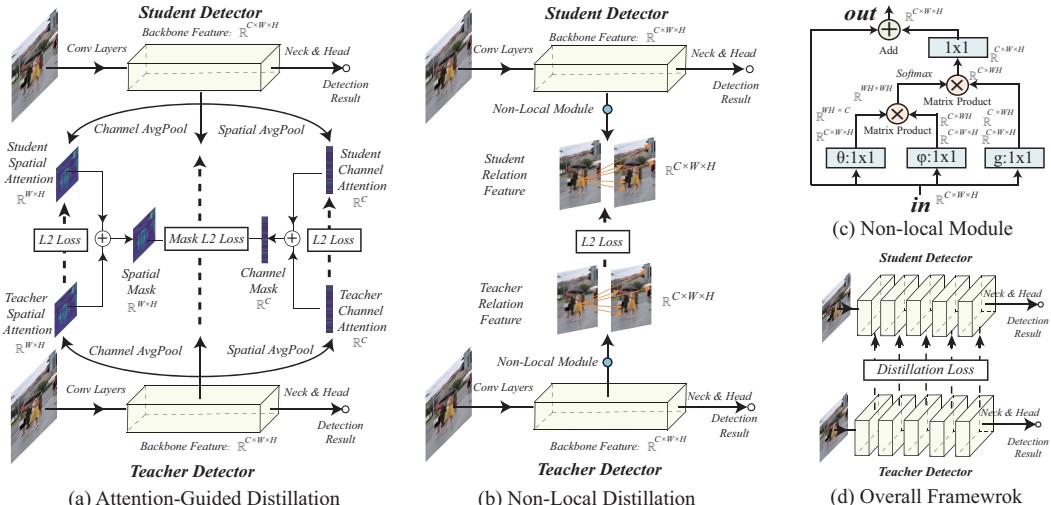

(a) Attention-Guided Distillation  (b) Non-Local Distillation  (c) Non-local Module  (d) Overall Framewrok

Figure 2: Details of our methods: (a) Attention-guided distillation generates the spatial and channel attention with average pooling in the channel and spatial dimension, respectively. Then, students are encouraged to mimic the attention of teachers. Besides, students are also trained to mimic the feature of teachers, which is masked by the attention of both students and teachers. (b) Non-local distillation captures the relation of pixels in an image with non-local modules. The relation information of teachers is learned by students with $L_2$ norm loss. (c) The architecture of non-local modules. '1x1' is convolution layer with 1x1 kernel. (d) Distillation loss is applied to backbone features with different resolutions. The detection head and neck are not involved in our methods.

face recognition (Ge et al., 2018), pretrained language model (Sanh et al., 2019; Xu et al., 2020), multi-exit networks training (Zhang et al., 2019b;a), model robustness (Zhang et al., 2020b) and so on. Hinton et al. (2014) first propose the concept of knowledge distillation where the students are trained to mimic the results after softmax layers of teachers. Then, abundant methods are proposed to transfer the knowledge in teacher's features (Romero et al., 2015) or the variants, such as attention (Zagoruyko & Komodakis, 2017; Hou et al., 2019), FSP (Yim et al., 2017), mutual information (Ahn et al., 2019), positive features (Heo et al., 2019), relation of samples in a batch (Park et al., 2019; Tung & Mori, 2019).

Improving the performance of object detection becomes a hot topic in knowledge distillation recently. Chen et al. (2017) design the first knowledge distillation method on object detection, which includes distillation loss on the backbone, the classification head and the regression head. Then, many researchers find that the imbalance between the foreground objects and background is a crucial problem in detection distillation. Instead of distilling the whole features of backbone networks, Li et al. (2017) only apply $L_2$ distillation loss to the features

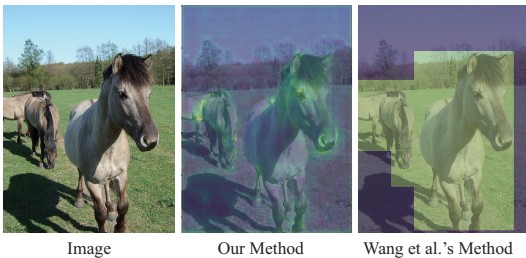

Image     Our Method     Wang et al.'s Method

Figure 3: Comparison between the proposed attention-guided distillation and other methods.

sampled by RPN. Bajestani & Yang (2020) propose the temporal knowledge distillation, which introduces a hyper-parameter to balance the distillation loss between the pixels of the foreground and background. Wang et al. (2019) propose the fine-grained feature imitation, which only distills the feature near object anchor locations. However, although these works have tried to distill only the pixels of foreground objects, they always reply on the annotation in groundtruth, anchors, and bounding boxes and thus can not be transferred to different kinds of detectors and tasks. In contrast, in our method, the pixels of foreground objects are found with attention mechanism, which can be easily generated from features. As a result, it can be directly utilized in all kinds of detectors without any modification. As shown in Figure 3, the difference between the previous mask-based detection distillation method (Wang et al., 2019) and our attention-guided distillation can be summarized as

follows (i) Our methods generate the mask with attention mechanism while they generate the mask with ground truth bounding boxes and anchor priors. (ii) The mask in our methods is a pixel-wise and fine-grained mask while the mask in their method is an object-wise and binary mask. (iii) The masks in our methods are composed of a spatial mask and a channel mask while they only have a spatial mask. More detailed comparison with related work can be found in Appendix.E.

## 3 METHODOLOGY

### 3.1 ATTENTION-GUIDED DISTILLATION

We use $A \in \mathbb{R}^{C,H,W}$ to denote the feature of the backbone in an object detection model, where $C, H, W$ denotes its channel number, height and width, respectively. Then, the generation of the spatial attention map and channel attention map is equivalent to finding the mapping function $\mathcal{G}^s : \mathbb{R}^{C,H,W} \to \mathbb{R}^{H,W}$ and $\mathcal{G}^c : \mathbb{R}^{C,H,W} \to \mathbb{R}^C$, respectively. Note that the superscripts $s$ and $c$ here are utilized to discriminate 'spatial' and 'channel'. Since the absolute value of each element in the feature implies its importance, we construct $\mathcal{G}^s$ by summing the absolute values across the channel dimension and construct $\mathcal{G}^c$ by summing the absolute values across the width and height dimension, which can be formulated as $\mathcal{G}^c(A) = \frac{1}{HW} \sum_H^{i=1} \sum_W^{j=1} |A_{\cdot,i,j}|$ and $\mathcal{G}^s(A) = \frac{1}{C} \sum_C^{k=1} |A_{k,\cdot,\cdot}|$, where $i, j, k$ denotes the $i_{th}, j_{th}, k_{th}$ slice of $A$ in the height, width, and channel dimension, respectively. Then, the spatial attention mask $M^s$ and the channel attention mask $M^c$ used in attention-guided distillation can be obtained by summing the attention maps from the teacher and the student detector, which can be formulated as $M^s = HW \cdot \text{softmax}((\mathcal{G}^s(A^{\mathcal{S}}) + \mathcal{G}^s(A^{\mathcal{T}}))/T), M^c = C \cdot \text{softmax}((\mathcal{G}^c(A^{\mathcal{S}}) + \mathcal{G}^c(A^{\mathcal{T}}))/T)$. Note that the superscripts $\mathcal{S}$ and $\mathcal{T}$ here are used to discriminate students and teachers. $T$ is a hyper-parameter in softmax introduced by Hinton et al. to adjust the distribution of elements in attention masks (see Figure 4). The attention-guided distillation loss $\mathcal{L}_{AGD}$ is composed of two components – attention transfer loss $\mathcal{L}_{AT}$ and attention-masked loss $\mathcal{L}_{AM}$. $\mathcal{L}_{AT}$ is utilized to encourage the student model to mimic the spatial and channel attention of the teacher model, which can be formulated as

$$\mathcal{L}_{AT} = \mathcal{L}_2(\mathcal{G}^s(A^{\mathcal{S}}), \mathcal{G}^s(A^{\mathcal{T}})) + \mathcal{L}_2(\mathcal{G}^c(A^{\mathcal{S}}), \mathcal{G}^c(A^{\mathcal{T}})). \tag{1}$$

$\mathcal{L}_{AM}$ is utilized to encourage the student to mimic the features of teacher models by a $\mathcal{L}_2$ norm loss masked by $M^s$ and $M^c$, which can be formulated as

$$\mathcal{L}_{AM} = \left( \sum_{k=1}^{C} \sum_{i=1}^{H} \sum_{j=1}^{W} (A_{k,i,j}^{\mathcal{T}} - A_{k,i,j}^{\mathcal{S}})^2 \cdot M_{i,j}^s \cdot M_k^c \right)^{\frac{1}{2}}. \tag{2}$$

### 3.2 NON-LOCAL DISTILLATION

Non-local module (Wang et al., 2018) is an effective method to improve the performance of neural networks by capturing the global relation information. In this paper, we apply non-local modules to capture the relation between pixels in an image, which can be formulated as $r_{i,j} = \frac{1}{WH} \sum_H^{i'=1} \sum_W^{j'=1} f(A_{\cdot,i,j}, A_{\cdot,i',j'}) g(A_{\cdot,i',j'})$, where $r$ denotes the obtained relation information. $i, j$ are the spatial indexes of an output position whose response is to be computed. $i', j'$ are the spatial indexes that enumerates all possible positions. $f$ is a pairwise function for computing the relation of two pixels and $g$ is an unary function for computing the representation of an individual pixel. Now, we can introduce the proposed non-local distillation loss $\mathcal{L}_{NLD}$ as the $\mathcal{L}_2$ loss between the relation information of the students and teachers, which can be formulated as $\mathcal{L}_{NLD} = \mathcal{L}_2(r^{\mathcal{S}}, r^{\mathcal{T}})$.

### 3.3 OVERALL LOSS FUNCTION

We introduce three hyper-parameters $\alpha, \beta, \gamma$ to balance different distillation loss in our methods. The overall distillation loss can be formulated as

$$\mathcal{L}_{Distill}(A^{\mathcal{T}}, A^{\mathcal{S}}) = \underbrace{\alpha \cdot \mathcal{L}_{AT} + \beta \cdot \mathcal{L}_{AM}}_{\text{Attention-guided distillation}} + \underbrace{\gamma \cdot \mathcal{L}_{NLD}}_{\text{Non-local distillation}}. \tag{3}$$

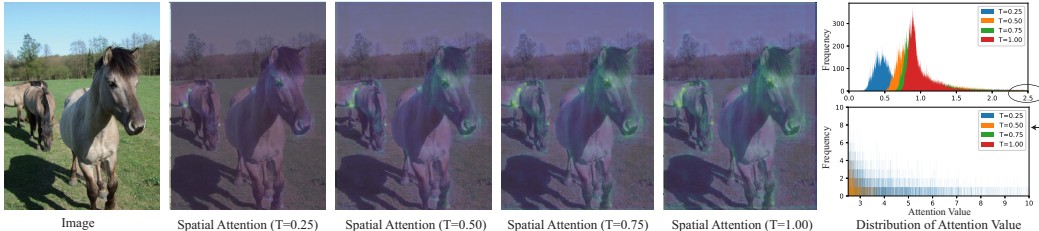

Figure 4: Visualization and distribution of the spatial attention with different $T$. With a smaller $T$, the pixels of high attention values are emphasized more in knowledge distillation.

The overall distillation loss is a model-agnostic loss, which can be added to the original training loss of any detection model directly. The sensitivity study of each hyper-parameter and the ablation study of each loss are shown in Figure 5 and Table 4, respectively.

# 4 EXPERIMENT

## 4.1 EXPERIMENTS SETTINGS

The proposed knowledge distillation method is evaluated on MS COCO2017, which is a large-scale dataset that contains over 120k images spanning 80 categories (Lin et al., 2014). The benchmark detection networks are composed of both two-stage detection models, including Faster RCNN (Ren et al., 2015), Cascade RCNN (Cai & Vasconcelos, 2019), Dynamic RCNN (Zhang et al., 2020a), Grid RCNN (Lu et al., 2019) and one-stage detection models, including the RetinaNet (Lin et al., 2017), Fsaf RetinaNet (Zhu et al., 2019). Besides, we also evaluate our methods on the Mask RCNN(He et al., 2017), Cascade Mask RCNN (Cai & Vasconcelos, 2019), and anchor-free models - RepPoints (Yang et al., 2019). We adopt the ResNet50 and ResNet101 (He et al., 2016) as the backbone network of each detection model. We pre-train the backbone model on ImageNet (Deng et al., 2009) and then finetune it on MS COCO2017. We have compared our methods with three kinds of object detection knowledge distillation methods (Chen et al., 2017; Wang et al., 2019; Heo et al., 2019). All the experiments in this paper are implemented with PyTorch (Paszke et al., 2019) with mmdetection2 framework (Chen et al., 2019). The reported fps is measured on one RTX 2080Ti GPU. We adopt the same hyper-parameters settings $\{\alpha = \gamma = 7 \times 10^{-5}, \beta = 4 \times 10^{-3}, T = 0.1\}$ for all the two-stage models and $\{\alpha = \gamma = 4 \times 10^{-4}, \beta = 2 \times 10^{-2}, T = 0.5\}$ for all the one-stage models. Cascade Mask RCNN with ResNeXt101 backbone is utilized as the teacher for all the two-stage students and RetinaNet with ResNeXt101 backbone is utilized as the teacher for all the one-stage students. Please refer to the codes in Github for more details.

## 4.2 EXPERIMENT RESULTS

In this section, we show the experiment results of the baseline detectors and our models in Table 1 and Table 2, and compare our methods with other three knowledge distillation methods in Table 3. It is observed that: (i) Consistent and significant AP boost can be observed on all the 9 kinds of detectors. On average, there are 2.9, 2.9, and 2.2 AP improvements on the two-stage, one-stage, and anchor-free detectors, respectively. (ii) With the proposed method, a student model with ResNet50 backbone can outperform the same model with ResNet101 backbone by 1.2 AP on average. (iii) On Mask RCNN related models, there are 2.3 improvements on bounding box AP and 2.0 improvements on mask AP on average respectively, indicating the proposed method can be utilized in not only object detection but also instance segmentation. (iv) Our methods achieve 2.2 higher AP than the second-best distillation method, on average. (v) There are 2.7 and 2.9 AP improvements on models with ResNet50 and ResNet101 backbones, respectively, indicating that deeper detectors benefit more from knowledge distillation.

Table 1: Experiments on MS COCO2017 with the proposed distillation method.

| Model | Backbone | AP | $AP_{50}$ | $AP_{75}$ | $AP_S$ | $AP_M$ | $AP_L$ | FPS | Params |
|-------|----------|-----|------|------|------|------|------|------|--------|
| Faster RCNN | ResNet50 | 38.4 | 59.0 | 42.0 | 21.5 | 42.1 | 50.3 | 18.1 | 43.57 |
| Our Faster RCNN | ResNet50 | 41.5 | 62.2 | 45.1 | 23.5 | 45.0 | 55.3 | 18.1 | 43.57 |
| Faster RCNN | ResNet101 | 39.8 | 60.1 | 43.3 | 22.5 | 43.6 | 52.8 | 14.2 | 62.57 |
| Our Faster RCNN | ResNet101 | 43.9 | 64.2 | 48.1 | 25.3 | 48.0 | 58.7 | 14.2 | 62.57 |
| Cascade RCNN | ResNet50 | 41.0 | 59.4 | 44.4 | 22.7 | 44.4 | 54.3 | 15.4 | 71.22 |
| Our Cascade RCNN | ResNet50 | 44.4 | 62.7 | 48.3 | 24.8 | 48.0 | 59.3 | 15.4 | 71.22 |
| Cascade RCNN | ResNet101 | 42.5 | 60.7 | 46.4 | 23.5 | 46.5 | 56.4 | 11.7 | 90.21 |
| Our Cascade RCNN | ResNet101 | 45.2 | 63.5 | 49.4 | 26.2 | 48.7 | 60.8 | 11.7 | 90.21 |
| Dynamic RCNN | ResNet50 | 39.8 | 58.3 | 43.2 | 23.0 | 42.8 | 52.4 | 18.1 | 43.57 |
| Our Dynamic RCNN | ResNet50 | 42.8 | 61.2 | 47.0 | 23.9 | 46.2 | 57.7 | 18.1 | 43.57 |
| Dynamic RCNN | ResNet101 | 41.2 | 59.7 | 45.3 | 24.0 | 44.9 | 54.3 | 14.2 | 62.57 |
| Our Dynamic RCNN | ResNet101 | 44.8 | 63.0 | 48.9 | 25.0 | 48.9 | 60.4 | 14.2 | 62.57 |
| Grid RCNN | ResNet50 | 40.4 | 58.4 | 43.6 | 22.8 | 43.9 | 53.3 | 14.0 | 66.37 |
| Our Grid RCNN | ResNet50 | 42.6 | 61.1 | 46.1 | 24.2 | 46.6 | 55.8 | 14.0 | 66.37 |
| Grid RCNN | ResNet101 | 41.6 | 59.8 | 45.0 | 23.7 | 45.7 | 54.7 | 11.0 | 85.36 |
| Our Grid RCNN | ResNet101 | 44.8 | 63.6 | 48.9 | 26.5 | 48.9 | 59.6 | 11.0 | 85.36 |
| RetinaNet | ResNet50 | 37.4 | 56.7 | 39.6 | 20.0 | 40.7 | 49.7 | 17.7 | 37.74 |
| Our RetinaNet | ResNet50 | 39.6 | 58.8 | 42.1 | 22.7 | 43.3 | 52.5 | 17.7 | 37.74 |
| RetinaNet | ResNet101 | 38.9 | 58.0 | 41.5 | 21.0 | 42.8 | 52.4 | 13.5 | 56.74 |
| Our RetinaNet | ResNet101 | 41.3 | 60.8 | 44.3 | 22.7 | 46.0 | 55.2 | 13.5 | 56.74 |
| Fsaf RetinaNet | ResNet50 | 37.8 | 56.8 | 39.8 | 20.4 | 41.1 | 48.8 | 20.0 | 36.19 |
| Our Fsaf RetinaNet | ResNet50 | 41.4 | 61.0 | 44.2 | 23.1 | 45.2 | 55.2 | 20.0 | 36.19 |
| Fsaf RetinaNet | ResNet101 | 39.3 | 58.6 | 42.1 | 22.1 | 43.4 | 51.2 | 15.0 | 55.19 |
| Our Fsaf RetinaNet | ResNet101 | 42.6 | 62.0 | 45.5 | 24.5 | 47.0 | 56.2 | 15.0 | 55.19 |
| RepPoints | ResNet50 | 38.6 | 59.6 | 41.6 | 22.5 | 42.2 | 50.4 | 18.2 | 36.62 |
| Our RepPoints | ResNet50 | 40.6 | 61.7 | 43.8 | 23.4 | 44.6 | 53.0 | 18.2 | 36.62 |
| RepPoints | ResNet101 | 40.5 | 61.3 | 43.5 | 23.4 | 44.7 | 53.2 | 13.2 | 55.62 |
| Our RepPoints | ResNet101 | 42.7 | 63.7 | 46.4 | 24.9 | 47.2 | 56.4 | 13.2 | 55.62 |

Table 2: Experiments on MS COCO2017 with the proposed distillation method on Mask RCNN.

| Model | Backbone | Bounding box AP | | | | Mask AP | | | |
|-------|----------|-----|--------|--------|--------|-----|--------|--------|--------|
| | | AP | $AP_S$ | $AP_M$ | $AP_L$ | AP | $AP_S$ | $AP_M$ | $AP_L$ |
| Mask RCNN | ResNet50 | 39.2 | 22.9 | 42.6 | 51.2 | 35.4 | 19.1 | 38.6 | 48.4 |
| Our mask RCNN | ResNet50 | 41.7 | 23.4 | 45.3 | 55.8 | 37.4 | 19.7 | 40.5 | 52.1 |
| Mask RCNN | ResNet101 | 40.8 | 23.0 | 45.0 | 54.1 | 36.6 | 19.2 | 40.2 | 50.5 |
| Our Mask RCNN | ResNet101 | 43.0 | 24.7 | 47.2 | 57.1 | 38.7 | 20.7 | 42.3 | 53.3 |
| Cascade Mask RCNN | ResNet50 | 41.9 | 23.2 | 44.9 | 55.9 | 36.5 | 18.9 | 39.2 | 50.7 |
| Our Cascade Mask RCNN | ResNet50 | 43.8 | 24.9 | 47.2 | 58.4 | 38.0 | 20.2 | 40.9 | 52.8 |
| Cascade Mask RCNN | ResNet101 | 42.9 | 24.4 | 46.5 | 57.0 | 37.3 | 19.7 | 40.6 | 51.5 |
| Our Cascade Mask RCNN | ResNet101 | 45.4 | 26.3 | 49.0 | 60.9 | 39.6 | 21.3 | 42.8 | 55.0 |

### 4.3 ABLATION STUDY AND SENSITIVITY STUDY

**Ablation study.** Table 4 shows the ablation study of the proposed attention-guided distillation ($\mathcal{L}_{AT}$ and $\mathcal{L}_{AM}$) and non-local distillation ($\mathcal{L}_{NLD}$). It is observed that: (i) Attention-guided distillation and non-local distillation lead to 2.8 and 1.4 AP improvements, respectively. (ii) $\mathcal{L}_{AT}$ and $\mathcal{L}_{AM}$ lead to 1.2 and 2.4 AP improvements respectively, indicating that most of the benefits of attention-guided distillation are obtained from the feature loss masked by the attention maps ($\mathcal{L}_{AM}$). (iii)

There are 3.1 AP improvements with the combination of attention-guided distillation and non-local distillation. These observations indicate that each distillation loss in our methods has their individual effectiveness and they can be utilized together to achieve better performance. We also give an ablation study to the spatial and channel attention in Appendix A.

**Sensitivity study on hyper-parameters.** Four hyper-parameters are introduced in this paper. $\alpha$, $\beta$, and $\gamma$ are utilized to balance the magnitude of different distillation loss and $T$ is utilized to adjust the distribution of attention masks. The hyper-parameter sensitivity study on MS COCO2017 with Faster RCNN (ResNet50 backbone) is introduced in Figure 5. It is observed that the worst hyper-parameters only lead to 0.3 AP drop compared with the highest AP, which is still 2.9 higher compared with the baseline model, indicating that our methods are not sensitive to the choice of hyper-parameters.

**Sensitivity study on the types of non-local modules.** There are four kinds of non-local modules, including Gaussian, embedded Gaussian, dot production, and concatenation. Table 5 shows the performance of our methods with different types of non-local modules. It is observed that the worst non-local type (Gaussian) is only 0.2 AP lower than the best non-local type (Embedded Gaussian and Concatenation), indicating our methods are not sensitive to the choice of non-local modules.

Table 3: Comparison between our methods and other distillation methods. Note that we don't compare our methods with Chen's and Wang's methods on RetinaNet because their methods can not be utilized in one-stage models. ResNet50 is utilized as backbone in these models.

| Model | AP | $AP_{50}$ | $AP_{75}$ | $AP_S$ | $AP_M$ | $AP_L$ |
|---|---|---|---|---|---|---|
| Faster RCNN | 38.4 | 59.0 | 42.0 | 21.5 | 42.1 | 50.3 |
| + Chen et al. Method | 38.7 | 59.0 | 42.1 | 22.0 | 41.9 | 51.0 |
| + Wang et al. Method | 39.1 | 59.8 | 42.8 | 22.2 | 42.9 | 51.1 |
| + Heo et al. Method | 38.9 | 60.1 | 42.6 | 21.8 | 42.7 | 50.7 |
| + Our Methods | **41.5** | **62.2** | **45.1** | **23.5** | **45.0** | **55.3** |
| Cascade RCNN | 41.0 | 59.4 | 44.4 | 22.7 | 44.4 | 54.3 |
| + Chen et al. Method | 41.6 | 62.1 | 45.7 | 23.9 | 45.8 | 54.7 |
| + Wang et al. Method | 42.1 | 62.7 | 46.1 | 24.4 | 46.4 | 55.6 |
| + Heo et al. Method | 41.7 | 62.3 | 45.7 | 23.9 | 45.7 | 55.3 |
| + Our Methods | **44.4** | **62.7** | **48.3** | **24.8** | **48.0** | **59.3** |
| RetinaNet | 37.4 | 56.7 | 39.6 | 20.0 | 40.7 | 49.7 |
| + Heo et al. Method | 37.8 | 58.3 | 41.1 | 21.6 | 41.2 | 48.3 |
| + Our Methods | **39.6** | **58.8** | **42.1** | **22.7** | **43.3** | **52.5** |

Table 4: Ablation study of the three distillation loss.

| Loss | $\mathcal{L}_{AT}$ | $\times$ | $\checkmark$ | $\times$ | $\times$ | $\checkmark$ | $\checkmark$ |
|---|---|---|---|---|---|---|---|
| | $\mathcal{L}_{AM}$ | $\times$ | $\times$ | $\checkmark$ | $\times$ | $\checkmark$ | $\checkmark$ |
| | $\mathcal{L}_{NLD}$ | $\times$ | $\times$ | $\times$ | $\checkmark$ | $\times$ | $\checkmark$ |
| Result | AP | 38.4 | 39.6 | 40.8 | 39.8 | 41.2 | **41.5** |
| | $AP_S$ | 21.5 | 22.7 | 22.8 | 22.7 | 23.0 | **23.5** |
| | $AP_M$ | 42.1 | 42.9 | 44.3 | 43.1 | 44.6 | **45.0** |
| | $AP_L$ | 50.3 | 52.5 | 54.3 | 52.3 | 55.3 | **55.3** |

Table 5: Results of different types of non-local modules on Faster RCNN (ResNet50 backbone).

| Non-Local Type | AP |
|---|---|
| Embedded Gaussian | **41.5** |
| Dot Production | 41.4 |
| Concatenation | **41.5** |
| Gaussian | 41.3 |

## 5 DISCUSSION

### 5.1 ANALYSIS ON THE BENEFITS OF KNOWLEDGE DISTILLATION

**Qualitative analysis.** Figure 6 shows the comparison of detection results between a baseline and a distilled detector. It is observed that: (i) Our methods improve the detection ability on small-objects. In the first three figures, the distilled model can correctly detect cars, the handbag, and the person in

the car, respectively. (ii) Our methods prevent models from generating multiple bounding boxes for the same object. In the last two figures, the baseline model generates multiple bounding boxes for the boat and the train while the distilled model avoids these errors.

**Analysis on the types of detection error.** We have analyzed the different types of detection errors in the baseline and distilled models in Figure 7. The number in the legend indicates AUC (area under the curve). It is observed that our distillation method leads to error reduction on all kinds of error. In brief, our methods can improve the ability of both localization and classification.

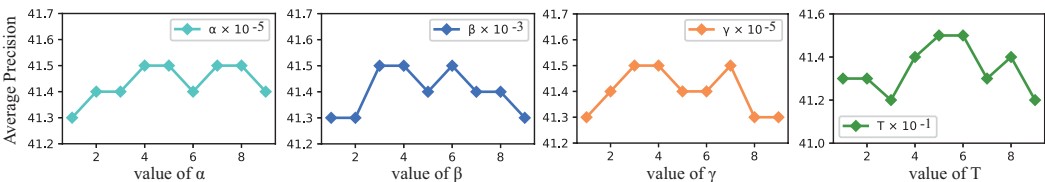

Figure 5: Hyper-parameter sensitivity study of $\alpha, \beta, \gamma, T$ with Faster RCNN on MS COCO2017.

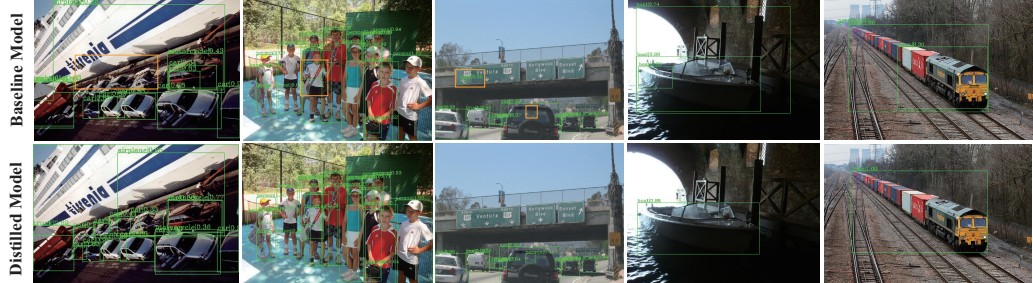

Figure 6: Qualitative analysis on MS COCO2017 with distilled and baseline Faster RCNN. We mark the undetected and wrongly detected objects of the baseline detector with orange boxes.

## 5.2 RELATION BETWEEN STUDENT DETECTORS AND TEACHER DETECTORS.

There is sufficient research focusing on the relation between students and teachers. Mirzadeh et al. (2019) and Cho & Hariharan (2019) show that a teacher with higher accuracy may not be the better teacher for knowledge distillation and sometimes a teacher with too high accuracy may harm the performance of students. Besides, Mobahi et al. (2020) and Yuan et al. (2019) show that the same model and even a model with lower accuracy than the student model can be utilized as the teacher model for knowledge distillation. However, all their experiments are conducted on image classification. In this section, we study whether these observations still hold in the task of object detection. As shown in Figure 8 , we conduct experiments on Faster RCNN (ResNet50 backbone) and Cascade RCNN (ResNet50 backbone) students with teacher models of different AP. It is observed that: (i) In all of our experiments, the student with a higher AP teacher always achieves higher AP. (ii) When the teacher has lower or the same AP as the student, there are very limited and even negative improvements with knowledge distillation. This observation indicates that the relation between students and teachers on object detection is opposite to that on image classification. Our experiment results suggest that *there is a strong positive correlation between the AP of students and teachers. A high AP teacher tends to improve the performance of students significantly.*

We think that the reason why a high AP teacher model is crucial in object detection but not very necessary in image classification is that object detection is a more challenging task. As a result, a weaker teacher model may introduce more negative influence on students, which prevents students from achieving higher AP. In contrast, on image classification, most of teacher models can achieve a very high training accuracy so they don't introduce so much error.

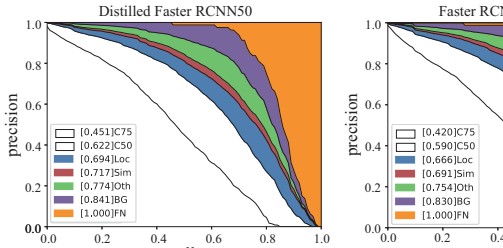 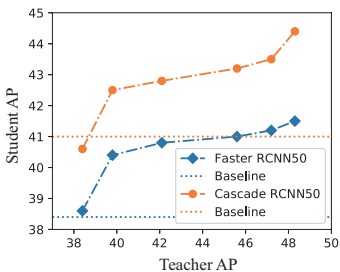

Figure 7: Distribution of error types on distilled and baseline Faster RCNN. **Loc** - Localization error; **Sim & Oth** - Classification error on similar & not similar classes; **BG** - False positive prediction fired on background. **FN** - False Negative prediction.

Figure 8: Relation between students and teachers on Faster RCNN and Cascade RCNN in MS COCO2017.

## 6 CONCLUSION

In this paper, we have proposed two knowledge distillation methods, including attention-guided distillation and the non-local distillation to improve the performance of object detection models. Attention-guided distillation manages to find the crucial pixels and channels from the whole feature map with attention mechanism and then enables the student to focus more on these crucial pixels and channels instead of the whole feature map. Non-local distillation enables students to learn not only the information of an individual pixel but also the relation between different pixels captured by the non-local modules. Experiments on 9 kinds of models including two-stage, one-stage, anchor-free and anchor-based models have been provided to evaluate our methods.

Besides, we have also given a study on the relation between students and teachers in object detection. Our experiments show that there is a strong positive correlation between the AP of teachers and students. A high AP teacher detector plays an essential role in knowledge distillation. This observation is much different from the previous conclusion in image classification, where a teacher model with very high accuracy may harm the performance of knowledge distillation. We hope that our result may call for more rethinking works on knowledge distillation in tasks except image classification.

## 7 ACKNOWLEDGEMENT

This work was partially supported by Institute for interdisciplinary Information Core Technology.

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

## A  ABLATION STUDY ON THE SPATIAL AND CHANNEL ATTENTION

Different from previous attention-based knowledge distillation methods, the attention-guided distillation in our methods uses not only spatial attention but also the channel attention. In this appendix, we have conducted an ablation study on the two kinds of attention with Faster RCNN (ResNet50) on MS COCO2017 to show their individual effectiveness.

It is observed that spatial attention and channel attention lead to 2.6 and 2.3 AP improvements, respectively. In contrast, the combination of the two kinds of attention leads to 2.8 AP improvements. These results indicate that both spatial and channel attention have their individual effectiveness and they can be utilized together to achieve better performance.

Table 6: Ablation study on the spatial attention and channel attention.

| Attention Type | Spatial | ✗ | ✓ | ✗ | ✓ |
|---|---|---|---|---|---|
| | Channel | ✗ | ✗ | ✓ | ✓ |
| Result | AP | 38.4 | 41.0 | 40.7 | **41.2** |
| | $AP_S$ | 21.5 | 22.7 | 22.9 | **23.0** |
| | $AP_M$ | 42.1 | 44.7 | 44.1 | **44.6** |
| | $AP_L$ | 50.3 | 54.2 | 54.1 | **55.3** |

## B ADAPTATION LAYERS IN KNOWLEDGE DISTILLATION

The adaptation layers in knowledge distillation are first proposed by Romero et al. (2015) to adjust the feature size of students and teachers. Then, recent research finds that the adaptation layers play an important role in improving the performance of students (Chen et al., 2017). In this paper, we adopt different kinds of adaptation layers for different distillation loss. Concretely, We adopt 1x1 convolutional layers for $\mathcal{L}_{AM}$ and $\mathcal{L}_{NLD}$, 3x3 convolutional layers for $\mathcal{L}_{AT}^{spatial}$, and fully connected layers for $\mathcal{L}_{AT}^{channel}$. Note that the adaptation layers are only utilized in the training period and they don't introduce additional computation and parameters.

## C EXPERIMENTS ON SMALLER BACKBONES

According to the insightful comments of the reviewers, we conduct a series of experiments on models with small backbones including ResNet18 and RegNet-800M, and compact detectors including Yolo v3 and SSD. As shown in Table 7, our methods also achieve significant AP improvements on these compact models. Note that more experiments with small backbones will be added in the camera ready version.

Table 7: Experiments on MS COCO2017 with our method on small backbones.

| Model | Backbone | AP | $AP_{50}$ | $AP_{75}$ | $AP_S$ | $AP_M$ | $AP_L$ | FPS | Params |
|---|---|---|---|---|---|---|---|---|---|
| Faster RCNN | ResNet18 | 34.6 | 55.0 | 37.1 | 19.3 | 36.9 | 45.9 | 28.1 | 30.57 |
| Our Faster RCNN | ResNet18 | 37.0 | 57.2 | 39.7 | 19.9 | 39.7 | 50.3 | 28.1 | 30.57 |
| Grid RCNN | ResNet18 | 36.6 | 54.2 | 39.7 | 20.1 | 39.8 | 48.2 | 26.7 | 66.37 |
| Our Grid RCNN | ResNet18 | 38.8 | 56.7 | 41.5 | 21.1 | 41.6 | 52.7 | 26.7 | 66.37 |
| RetinaNet | ResNet18 | 33.4 | 51.8 | 35.1 | 16.9 | 35.6 | 44.9 | 25.8 | 23.30 |
| Our RetinaNet | ResNet18 | 35.9 | 54.4 | 38.0 | 17.9 | 39.1 | 49.4 | 25.8 | 23.30 |
| RetinaNet | RegNet-800M | 35.6 | 54.7 | 37.7 | 19.7 | 39.0 | 47.8 | 22.4 | 19.27 |
| Our RetinaNet | RegNet-800M | 38.4 | 57.4 | 40.7 | 21.4 | 42.0 | 52.3 | 22.4 | 19.27 |
| Yolo v3 | DarkNet53 | 33.4 | 56.3 | 35.2 | 19.5 | 36.4 | 43.6 | 42.2 | 61.95 |
| Our Yolo v3 | DarkNet53 | 35.8 | 58.2 | 38.1 | 21.2 | 39.0 | 45.6 | 42.2 | 61.95 |
| SSD | VGG16 | 29.4 | 49.3 | 31.0 | 11.7 | 34.1 | 44.9 | 26.1 | 38.08 |
| Our SSD | VGG16 | 31.2 | 52.1 | 32.8 | 12.6 | 37.4 | 46.2 | 26.1 | 38.08 |

## D EXPERIMENTS ON CITYSCAPES

According to the insightful comments of the reviewers, as shown in Table 8, we conduct a series of experiments to show the effectiveness of our method on Cityscapes. Note that more experiments on Cityscapes will be added in the camera ready version.

Table 8: Experiments on Cityscapes with our method.

| Model | Backbone | Box AP | Mask AP |
|---|---|---|---|
| Faster RCNN | ResNet50 | 40.3 | N/A |
| Our Faster RCNN | ResNet50 | 43.5 | N/A |
| Mask RCNN | ResNet50 | 41.0 | 35.8 |
| Our Mask RCNN | ResNet50 | 43.0 | 37.5 |

# E    COMPARISION WITH RELATED WORK

**Comparison on Methodology and Application.** Feature distillation is utilized in all the five methods. However, Chen et al. distill not only the feature, but also the classification logits and bounding box regression results, which has limited their application scenes in one stage and anchor-free models. Li et al. and Wang et al. distill the features in the regions of proposals and near object anchor locations, respectively. As a result, their methods reply on the supervision of anchors and groundtruths and can't be utilized in one stage and anchor free models. Bajestani & Yang is utilized for active perception on video, which can not be utilized in image-based detection. In contrast, in our method, the attention mask and relation information can be easily generated from the backbone features, which has no requirements on groundtruths, anchors and proposals. As a result, it can be easily used in different kinds of models and tasks without any modification.

Table 9: Comparision on methodology and application.

| Method | Feature | Classify | Regress | Relation | One-Stage | Two-Stage | Anchor-based | Anchor-free | Segment. |
|---|---|---|---|---|---|---|---|---|---|
| Our Method | ✓ | ✗ | ✗ | ✓ | ✓ | ✓ | ✓ | ✓ | ✓ |
| Chen et al. | ✓ | ✓ | ✓ | ✗ | ✓ | ✓ | ✓ | ✗ | ✗ |
| Li et al. | ✓ | ✗ | ✗ | ✗ | ✗ | ✓ | ✓ | ✗ | ✗ |
| Wang et al. | ✓ | ✗ | ✗ | ✗ | ✗ | ✓ | ✓ | ✗ | ✗ |
| Bajestani & Yang | ✓ | ✗ | ✗ | ✗ | ✓ | ✓ | ✓ | ✗ | ✗ |

**Comparison on Motivation.** Chen et al.'s method is a direct application of knowledge distillation on object detection. The other three methods and our method are motivated by the imbalance between foreground and background piexles and these methods try to address this issue by reweighting the distillation loss. Besides, our method is also motivated by the effect of the relation among pixels in an image, which is ignored by the other methods.

