# OpenReview forum: "Improve Object Detection with Feature-based Knowledge Distillation: Towards Accurate and Efficient Detectors"
_ICLR.cc/2021/Conference — ICLR 2021 Poster_

### Official Review · AnonReviewer4 · 2020-10-27
**This paper proposes two knowledge distillation methods to improve the performance of object detection models. With respect to other methods it proposes local distillation as well as attention-guided distillation to improve the knowledge transfer between teacher and student.**

**Rating:** 6
**Confidence:** 4

**Review:**

Pros:

- The different attention techniques seem to consistently improve object detectors across different models.
- The ablation studies are important in showing the advantage and impact of each proposed module.
- Please clarify if the student models start from random weights or are initialized after the teacher training. In this regard to what extend is the approach is transferable to students with random weights?

Cons:

- The authors only show results when training a network of the same structure with the added modules, in an attempt to boost accuracy. However, an important use of knowledge of knowledge distillation is to train smaller models, and such results are missing from the paper. Overall, the results in the paper somewhat justify the "accurate" part of the title but not the "efficient".

- The importance of high AP is a bit overstated. It is important to demonstrate the different behavior between classification and detection networks, however, the same improvement is observed irrespective of the teacher AP performance.

- The structure of the paper can be improved. The related work section for example can be moved earlier in the paper to give the bigger picture and the position of this work with respect to the literature.

- For a more complete comparison YOLO and SSD networks should be included.

- In the qualitative analysis observation (ii) states that the proposed methods leads to single box per object compared to the baselines. In this case do the baselines use non-maximum suppression? This is a standard post processing technique that alleviates this problem. No explanation is given on why the proposed method leads to this behavior.

The authors have made clear some of my concerns and made revisions accordingly. Thus I am in a position now to recommend this paper, thus I update my initial recommendation from 5 to 6.

---

> ### Author Response · Authors · 2020-11-16
> **Thanks for your insightful comments! We have added experiments with small backbones, experiments on Yolo and SSD, changed the structured of the paper and added illustration on qualitative analysis**
>
> **We sincerely appreciate your detailed and thoughtful reviews and hope that our response can address your questions.**
>
> ---
>
> **Response to Pros.3. About weights initialization.**
> **The weights of teachers are not used in the initialization of students**. We initialize the weights of the backbone in the students with pre-trained models in ImageNet. The other weights of students are initialized randomly with Xavier initialization [1].
>
> [1] Glorot X, Bengio Y. Understanding the difficulty of training deep feedforward neural networks[C] Proceedings of the thirteenth international conference on artificial intelligence and statistics
>
>
>
> **Question1. Experiments on smaller models** & **Question4. Experiments on Yolo and SSD**
> Thanks for your suggestions on conducting experiments on smaller models. The following table shows the experiment results on Faster R-CNN and RetianNet with smaller backbones (ResNet18 and RegNet - 800M). Besides, we also evaluate our methods on compact detectors, including Yolo v3 and SSD. It is observed that our method also achieves significant AP improvements on smaller models.
>
> By replacing the large detector with a small detector trained with distillation, we can achieve significant compression and acceleration with trivial AP decrement, or even AP boost. For example, by replacing a baseline RetinaNet (ResNet50 backbone) with a distilled RetinaNet (RegNet-800M backbone), we can achieve 1.3 X acceleration,  2.0 X compression, and 1.0 AP improvements. We will report more experiments on small backbones in the camera-ready vision.
>
>
>
> |Model| Backbone| Distill?| AP (MS COCO)|Inference Time (FPS)|Params (M)|
> |------------|-----------|--------|----|----|----------|
> |Faster R-CNN|ResNet18|×|34.6|28.1|30.57|
> |Faster R-CNN|ResNet18|√|37.0|28.1|30.57|
> |Grid R-CNN|ResNet18|x|36.6|26.7|66.37|
> |Grid R-CNN|ResNet18|√|38.8|26.7|66.37|
> |RetinaNet|ResNet18|×|33.4|25.8|23.30|
> |RetinaNet|ResNet18|√|35.9|25.8|23.30|
> |RetinaNet|RegNet-800M|×|35.6|22.4|19.27|
> |RetinaNet|RegNet-800M|√|38.4|22.4|19.27|
> |Yolo v3|DarkNet53|×|33.4|42.2|61.95|
> |Yolo v3|DarkNet53|√|35.8|42.2|61.95|
> |SSD|VGG16|×|29.4|26.1|38.08|
> |SSD|VGG16|√|31.2|26.1|38.08|
>
>
>
> **Question2. About the evaluation measurement and Student-Teacher relation.**
>
> **Evaluation Measurement**. We agree that AP is not a perfect evaluation measurement for object detection. To address this concern, we additionally report the AR (average recall) on COCO in the following table. It is observed that our distillation method also leads to significant AR improvements. Note that recall is another important evaluation metric in object detection, which measures the probability of ground truth objects being correctly detected. More results can be found in the log files in our supplementary material. We are glad to evaluate any other evaluation measurement which is required by you.
>
> |Model| Backbone|Distill?|AP (Average Precision)| AR (Average Recall)| AR small  | AR medium  | AR large  |
> |------------ | -------- | --------- |----|----|----|----|----|
> |Faster R-CNN|ResNet50|× |38.4|52.0|32.6|55.8|66.1|
> |Faster R-CNN|ResNet50|√|41.5|54.4 |34.0|58.2|69.9|
> |RetinaNet|ResNet50|×|37.4|53.9|33.1|57.7|70.2|
> |RetinaNet|ResNet50|√ |39.6|56.1|36.8|60.0|72.1|
> |RepPoints|ResNet50|×|38.6|55.1|34.9|59.4|70.3|
> |RepPoints|ResNet50|√|40.6|56.9|37.3|60.9|71.4|
>
> **Student-Teacher relation**. As shown in Figure.7 and Section 4.2, our experiments show that in object detection, there is a strong positive correlation between the AP of teachers and students.
>
>
>
> **Question3. The structure of the paper.**
> Thanks for your advice on the structure of the paper. We have moved the related work as the second section in the rebuttal revision according to your advice.
>
> **Question4. Results on Yolo v3 and SSD.**
> Please refer to our above response to **Q1&Q4**. Note that these results have also been added in our rebuttal revision (Appendix.C).
>
>
> **Question5. About NMS in qualitative analysis.**
>
> **If NMS is used?**  Yes, we have used non-maximum suppression in both baseline and distilled models in our qualitative analysis. Note that although NMS can protect detectors from generating multiple bounding boxes for the same object, *it can not solve this problem perfectly*.
>
> **Why does knowledge distillation help?**  Since knowledge distillation provides better backbone features to the RPN layer, the RPN layer can predict better bounding boxes, which helps to address the "multiple boxes - single object" problem.
>
> **Reproduce qualitative analysis**.  To address your confusion, we also provide the script to reproduce the qualitative analysis as follows.
>
> ```
> #	Step-1: Install the mmdetection framework.
> #	Step-2: Download the pre-trained faster r-cnn from the model zoo and COCO dataset.
> #	Step-3: Conduct qualitative analysis by running the following scripts.
> python tools/test.py \
>     configs/faster_rcnn/faster_rcnn_r50_fpn_1x.py \
>     checkpoints/"model_file_name" \
>     --show-dir "output_folder_name"
> ```

---

### Official Review · AnonReviewer1 · 2020-10-27
**Good improvements in empirical results, experimental analysis and discussion with related work can be improved to provide further insights.**

**Rating:** 6
**Confidence:** 4

**Review:**

The paper proposes a knowledge distillation method for object detection. In particular, the technical contribution is mainly two-fold: attention-guided distillation module and non-local distillation module, as shown in (a) and (b) of Figure 2. The proposed modules provide consistent improvements in detection MAP across different architectures.

On the positive side, I believe the paper has the following merits:
- The proposed modules are well-motivated, and seem to provide a consistent boost in different architectures.
- The authors provide very detailed information to reproduce the method. Besides, I also appreciate the authors have provided the code in the supplementary material.
- The observation that a high-AP teacher is important for distillation is quite intriguing.

Overall I believe the paper has made a decent contribution, but the following aspects can be improved:
- As mentioned by the authors, distillation is discussed for object detection in several works (n (Chen et al., 2017; Li et al., 2017; Wang et al., 2019; Bajestani & Yang, 2020). I believe it would be helpful to summarize the key difference/similarity wrt to the previous work, thus to provide a better understanding of the relation in a larger context.
- it's not fully clear to me what the attention in attention-guided distillation captures, would the authors provide further experimental analysis? and what are the failure cases of attention-guided distillation?
- it would be helpful to provide results on another dataset(e.g. cityscapes) to confirm the effectiveness in a different detection setting.

---
After reading the authors' response and other reviews. I still believe the paper has made a good contribution thus I would stick with my original rating.

---

> ### Author Response · Authors · 2020-11-16
> **Thanks for your insightful comments! We have added more literature review, Cityscapes experiments and explanation of attention.**
>
> **We sincerely appreciate your detailed and thoughtful reviews and hope that our response can address your questions.**
>
> ---
>
> **Question1. Summary of key difference and similarity with previous works**
>
> Thanks for your advice on the literature review. We have summarized the key difference and similarity of our method and previous works in the following table and added them in our rebuttal revision.
>
> |         Method         | Feature Distillation | Classification Distillation | Regression Distillation | Relation Distillation | One Stage | Two Stage | Anchor Based | Anchor Free | Instance Segment |
> | :--------------------: | :------------------: | :-------------------------: | :---------------------: | :-------------------: | :-------: | :-------: | :----------: | :---------: | :--------------: |
> |       Our Method       |√|×| ×|√|√ |√  |√ | √ | √ |
> |   Chen et al., 2017    | √|√|√|× |  × |√ |√ |×| × |
> |    Li et al., 2017     |√ |×|× | × | ×|√  | √  |× |×|
> |    Wang et al.2019     |√  | × |× |× | ×  | √  |      √       |      ×      |  ×    |
> | Bajestani & Yang, 2020 |√| ×| × |× | √ |     √     | √  |      ×      |  ×     |
>
>
>
>
> **Comparison on Methodology and Application**
>
> Knowledge distillation on the backbone features is utilized in all these methods. However, Chen et al distill not only the feature but also the classification logits and bounding box regression results, which has limited their application scenes in one stage and anchor-free models. Li et al. and Wang et al. distill the features in the regions of proposals and near object anchor locations, respectively. As a result, their methods rely on the supervision of anchors and ground truths and can't be utilized in one stage and anchor free models. Bajestani&Yang's method is utilized for active perception on video data, which can not be utilized in image-based detection.
>
> In contrast, in our method, the attention mask and relation information can be easily generated from the backbone features, which has no requirements on ground truths, anchors or proposals. As a result, it can be easily used in different kinds of models and tasks without any modification.
>
> **Comparison on Motivation**
>
> Chen et al.'s method is a direct application of knowledge distillation on object detection. The other three methods and our method are motivated by the imbalance between foreground and background pixels and these methods try to address this issue by reweighting the distillation loss. Besides, our method is also motivated by the effect of the relation among pixels in an image, which is ignored by the other methods.
>
> | Method | Motivation 1: Imbalance between foreground and background | Motivation 2: Lack of Relation Distillation |
> | :--------------------: | :-------------------------------------------------------: | :-----------------------------------------: |
> |Our Method |√|√|
> |Chen et al., 2017|×|√|
> |Li et al., 2017 |√|×|
> |Wang et al.2019 |√|×|
> |Bajestani & Yang, 2020 |√| ×|
>
>
> **Question2. About the attention in attention-guided distillation.**
>
> **What does it capture?** The attention captures the crucial pixels in the feature map. For example, as shown in Figure.3, most of the pixels with high attention value focus on the head, legs, and eyes of the horses while the pixels of the background have much lower attention value.
>
> The attention here is generated by summing the absolute value of features in the channel dimension. As a result, the pixel which has a larger absolute value will have a larger attention value. Note that this observation has been also utilized and discussed in [1-2]. The visualization and distribution of attention have been provided in Figure3.
>
> **Failure case.** The attention in attention-guided distillation may fail in a few specific cases. For example, in Figure3, some pixels in the tree also have high attention value. But these failure attention cases are much less than the correct cases and thus they will not introduce much negative influence during knowledge distillation.
>
> [1] Sergey Zagoruyko et al. Paying more attention and Nikos Komodakis. attention: Improving the performance of convolutional neural networks via attention transfer. ICLR, 2017.
>
> [2] Yuenan Hou et al. Learning lightweight lane detection cnns by self attention distillation. ICCV2019.
>
> Note that these two papers have also been cited in our paper.
>
> **Question3.Results on Cityscapes.**
> Thanks for your advice on conducting experiments on Cityscapes. As shown in the table, we have conducted experiments with Faster R-CNN and Mask R-CNN on Cityscapes. It is observed that our method also leads to significant AP improvements in Cityscapes. We will add more Cityscapes experiments in the camera-ready vision.
>
> | Model| Backbone | Distill ? | Box AP | Mask AP |
> | ------------ | -------- | --------- | ------ | ------- |
> | Faster R-CNN|ResNet50 |×| 40.3| N/A|
> | Faster R-CNN|ResNet50 |√| 43.5| N/A|
> | Mask R-CNN|ResNet50 |×| 41.0| 35.8|
> | Mask R-CNN|ResNet50 |√| 43.0 | 37.5|

---

### Official Review · AnonReviewer2 · 2020-10-29

**Rating:** 7
**Confidence:** 5

**Review:**

This paper explores the knowledge distillation problem in object detection. It claims that the failure of knowledge distillation in object detection is mainly caused by the imbalance between pixels of foreground and background, and the relation distillation between different pixels. The authors then propose non-local distillation to tackle the problem. Extensive experiments are conducted on MS COCO and verify the effectiveness of the proposed method.

1. Strengthens

a. This paper explores the knowledge distillation problem in object detection and gives promising results and conclusions.

b. The results with different detectors greatly prove the effectiveness of the method.

c. It is good that the distillation method does not bring any extra costs during the inference time.

2. Weaknesses

a. I recommend the authors do experiments on much smaller models. For example, in real cases, mimicking ResNet50 with larger models is meaningless. It would be great if we can observe large improvements when mimicking a small backbone with ResNet-50 or even larger models. These conclusions and experiments are crucial for the applications of the method.

b. The English writing of this paper can be improved.

c. The names of the models in this paper are not proper. It is weird for me to call Faster RCNN101. Generally, Faster R-CNN is Faster R-CNN, and ResNet is ResNet. You can use any backbones with Faster R-CNN. Please define it as Faster R-CNN with ResNet101.

As stated comments, I think this paper is good for knowledge distillation in object detection and publishable, but encourages the authors improve the draft according to the weaknesses.

---

> ### Author Response · Authors · 2020-11-16
> **Thanks for your insightful comments! We have added experiments on small backbones and corrected the name of models according to your advice.**
>
> **We sincerely appreciate your detailed and thoughtful reviews and hope that our response can address your concerns.**
>
> **Question1. Experiments on smaller models**
> Thanks for your suggestions on conducting experiments on smaller models. The following table shows the experiment results on Faster R-CNN and RetianNet with smaller backbones (ResNet18 and RegNet - 800M) on MS COCO2017. Besides, we also evaluate our methods on compact detectors, including Yolo v3 and SSD. It is observed that our method also achieves significant AP improvements on smaller models. We will report more experiments on small backbones in the camera-ready vision.
>
>
> | Model        | Backbone    | Distill? | AP (MS COCO)   | Inference Time (FPS)  | Params (M) |
> | ------------ | ----------- | -------- | ---- | ---- | ---------- |
> | Faster R-CNN | ResNet18    | ×        | 34.6  | 28.1 | 30.57      |
> | Faster R-CNN | ResNet18    | √        | 37.0 | 28.1 | 30.57      |
> | Grid R-CNN   | ResNet18    | x        | 36.6 | 26.7 | 66.37      |
> | Grid R-CNN   | ResNet18    | √        | 38.8 | 26.7 | 66.37      |
> | RetinaNet    | ResNet18    | ×        | 33.4 | 25.8 | 23.30      |
> | RetinaNet    | ResNet18    | √        | 35.9 | 25.8 | 23.30      |
> | RetinaNet    | RegNet-800M | ×        | 35.6 | 22.4 | 19.27      |
> | RetinaNet    | RegNet-800M | √        | 38.4 | 22.4 | 19.27      |
> | Yolo v3      | DarkNet53   | ×        | 33.4 | 42.2 | 61.95      |
> | Yolo v3      | DarkNet53   | √        | 35.8 | 42.2 | 61.95      |
> | SSD          | VGG16       | ×        | 29.4 | 26.1 | 38.08      |
> | SSD          | VGG16       | √        | 31.2 | 26.1 | 38.08      |
>
>
>
> **Question2. Paper writing**
>
> Thanks for your suggestion, we will carefully revise the manuscript to make it easier to understand.
>
> **Question3. The name of models**
>
> Thanks for your suggestions on the names of models. We have addressed these issues in the rebuttal revision according to your advice.

---

### Decision · Program_Chairs · 2021-01-07
**Final Decision**

**Decision:**

Accept (Poster)

**Comment:**

After the rebuttal stage, all reviewers lean positive (in final scores and/or in comments during the discussion phase). The AC found no reason to disagree. The benefit of the proposed method is demonstrated in many diverse settings, and the authors argue novelty in that no prior work addresses both fg/bg imbalance and relation distillation.